# Development and Application of an Intelligent Plant Protection Monitoring System

Shubo Wang [1,2,3], Peng Qi [1,2,3], Wei Zhang [4] and Xiongkui He [1,2,3,*]

1  Centre for Chemicals Application Technology, China Agricultural University, Beijing 100193, China; wangshubo@cau.edu.cn (S.W.); qp@cau.edu.cn (P.Q.)
2  College of Agricultural Unmanned System, China Agricultural University, Beijing 100193, China
3  College of Science, China Agricultural University, Beijing 100193, China
4  Anhui Zhongke Intelligent Perception Industrial Technology Research Institute Co., Ltd., Wuhu 241000, China; zhangw@zkzngz.com
*  Correspondence: xiongkui@cau.edu.cn

**Abstract:** Facing the need of modern agriculture to accurately grasp the information of farmland diseases and pests, this paper proposes an intelligent plant protection system. The system is composed of a wireless lens, temperature and humidity sensor, intelligent information terminal, and probe rod to realize the collection of plant images and meteorological information. At the same time, a software based on the mobile terminal and the computer terminal was developed. The plant images and meteorological data are transmitted to the server through Wi-Fi transmission. Combined with the expert knowledge model, a solution is generated, and the user can identify the current diseases and pests and obtain solutions at any time. The system can remotely and automatically monitor and warn of mainstream diseases and pests of field crops such as rice and wheat and provide support for fine plant protection management.

**Keywords:** farmland information; disease and pest identification; temperature and humidity acquisition; mobile monitoring





## 1. Introduction

Diseases, pests, and weeds are the great enemies of crop growth. They occur in the entire growth period of crops, which can cause a large reduction in crop production [1,2]. The application of chemical pesticides can greatly reduce agricultural losses, but the "three causes" caused by the use of chemical pesticides (referring to the mutagenic, carcinogenic, and teratogenic effects of pesticides on higher animals), pesticide residues, environmental pollution, and other negative effects are becoming increasingly prominent. In addition to experienced experts, for non-professional researchers, especially farmers, identifying crop diseases through picture comparison or text description often leads to human judgment errors [3,4]. In this way, it is difficult to accurately and timely apply pesticides, resulting in a large reduction in crop production. More and more signs also show that the contradiction between the increasing demand for pest identification and the relatively few plant protection experts has become increasingly serious [5]. It is of great significance to find new ways to solve the problem of diseases and pests identification.

With the rapid development of digital image processing, the increasingly wide application of computer vision, and the maturity of various pattern recognition technologies, the purpose of intelligent recognition can be achieved by processing the images of crop diseases, pests, and weeds and extracting characteristic parameters. Many scholars have studied the intelligent perception of diseases, pests, and weeds. In terms of methodology, machine learning models for agricultural monitoring are widely developed and used at this stage [6,7]. Nagasubramanian et al. [8] developed an analytical statistics model for plant growth and disease patterns based on a convolutional neural network (CNN) and

support vector machine (SVM). This framework supported irrigation, nutrition planning, and environmental compliance. Albanese et al. [9] presented an embedded system enhanced with ML functionalities, ensuring the continuous detection of pest infestation inside fruit orchards. In their work, three different machine learning (ML) algorithms had been trained and deployed. Mishra et al. [10] proposed a multi-node plant disease-monitoring strategy, collected plant images through nodes, and used the proposed sine cosine algorithm based on the rider neural network to identify diseases. The large-scale monitoring of farmland could be realized by using a multi-node method. Liu et al. [11] proposed a novel, deep learning-based automatic approach using hybrid and local activated features for pest monitoring. The authors tested their algorithm in seven years of large-scale pest data set containing 88,600 images (16 types of pests) with 582,100 pests and achieved good results.

In terms of hardware, there are ground-monitoring platforms and air-monitoring platforms [12–15]. For ground-monitoring platforms, Trilles et al. [16] developed a low-cost sensorized platform, capable of monitoring meteorological phenomena following the Internet of Things paradigm, with the goal to apply an alert disease model on the cultivation of the vine. However, the platform could only carry out meteorological monitoring, and it was difficult to support the implementation of intelligent agriculture due to insufficient information. Methun et al. [17] proposed an efficient carrot disease identification and classification method using a deep learning approach, especially VGG16, VGG19, MobileNet, and Inception v3. The accuracy achieved by Inception v3 was 97.4%. Udutalapally et al. [18] developed a disease prediction system disease prediction by trained a convolutional neural network (CNN) model. Considering the power problem of field environment, the solar sensor node was used for power supply. Khattab et al. [19] collected the environmental and soil information from a wireless sensor network installed in the planted area. It also provided an expert system that sent warning messages to the users before the outbreak of a disease. Mekala et al. [20] proposed a cloud-enabled CLAY-MIST measurement (CMM) index based on temperature and relative humidity to assess the comfort levels of a crop.

For an air-monitoring platform, He et al. [21] used multiangle remote sensing and hyperspectral remote sensing to monitor wheat powdery mildew (WPM). Dong et al. [22] developed an automatic system based on the Web GIS platform. At the same time, the author used it to predict wheat stripe rust and East Asian migratory locust and achieved good accuracy. Kim et al. [23] proposed an onion field monitoring system which composed of a Pan Tilt Zoom (PTZ) camera, a motor system, wireless translator, and image-logging module. Through pixel-level classification and location, six categories of disease recognition were realized.

In addition to ground systems and air systems, it is also a trend to integrate them for collaborative monitoring. Gao et al. [24] realized the joint detection of environmental information by ground sensors and UAVs. IoT and UAV could monitor the incidence of crop diseases and pests from the ground micro and air macro perspectives, respectively. Zhang et al. [25] proposed to use high-resolution remote sensing data to identify infected trees for controlling pine wilt disease (PWD). By processing the captured spectral, temporal, and spatial features, this paper effectively distinguished the withered pine from other easily confused objects.

In view of the above research status, as shown in the Figure 1, this paper has developed a variety of crop diseases and pests monitoring systems, taking into account meteorological conditions, plant images, and expert systems. The lower half of Figure 1 shows the acquisition of images and meteorological data. The wireless lens collects the crop images, and the Bluetooth temperature and humidity sensor collects the environmental information in real time. The upper half of Figure 1 shows the data upload and discrimination results. The obtained images are uploaded to the cloud expert system to realize the classification of diseases and pests. Additionally, the expert system can release prevention and control suggestions for users. The main contributions of this work are:

(1)　A complete set of monitoring systems is constructed, including the software and hardware of the system.

(2)　The system can collect meteorological data and plant images at the same time and realize the identification and counting of pests and diseases based on Yolo v3. A timely solution can be given through the expert system.

The rest of the paper is organized as follows. Section 2 introduces the diseases and pests identification. In Section 3, the hardware system is developed and applied. In Section 4, software systems based on terminal and computer are developed. Section 5 describes the detailed acquisition method and representative results. Section 6 concludes the paper.

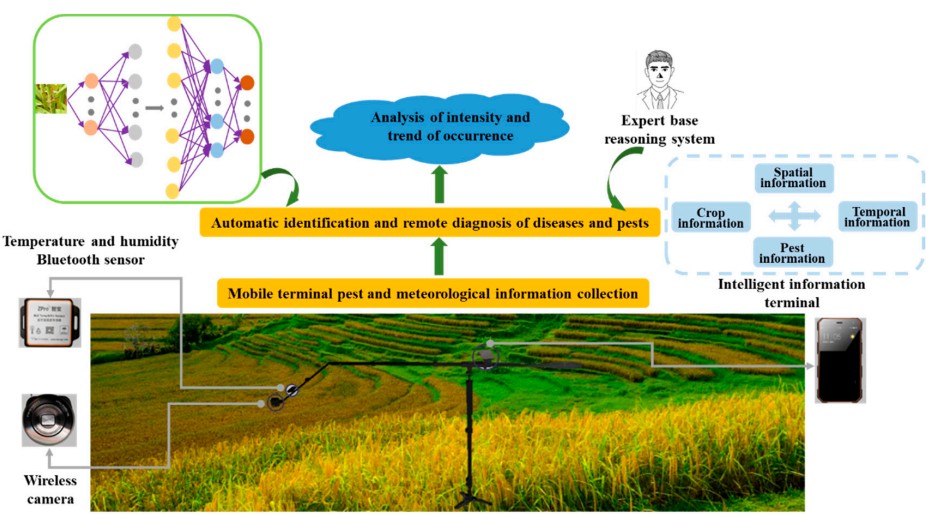

**Figure 1.** Overview of intelligent plant protection monitoring system.

## 2. Diseases and Pests Identification

### 2.1. Image Acquisition and Coding of Diseases and Pests in Macro Mode

As shown in Table 1, the names of diseases and pests are set to a number with a length of five. The combination of the first and second bits indicates the crop type; the third bit indicates diseases or pests; and the combination of the fourth and fifth bits indicates the specific kinds of diseases and pests.

**Table 1.** Coding table of diseases and pests with different crops.

| Crop | Pests and Diseases | Specific Diseases and Pests | Coding |
| --- | --- | --- | --- |
| 11: Wheat | 1: Pest | 11: Wheat spider | 11111 |
| | | 21: Aphid, wheat long tube aphid or wheat binary aphid (green) | 11121 |
| | | 22: Aphid, rhopalosiphus (black) | 11122 |
| | | 23: Aphids (red) | 11123 |
| | | 31: Slime worm | 11131 |
| | 2: Disease | 11: Normal wheat ears | 11211 |
| | | 12: Scab | 11212 |
| | | 21: Powdery mildew | 11221 |
| | | 31: Wheat leaf rust | 11231 |
| | | 41: Wheat stripe rust | 11241 |

**Table 1.** *Cont.*

| 12: Rice | 1: Pest | 11: Rice planthopper | 12111 |
|---|---|---|---|
| | | 21: Normal rice leaves | 12121 |
| | | 22: Rice-leaf roller | 12122 |
| | | 31: Striped rice borer | 12131 |
| | 2: Disease | 11: Sheath blight | 12211 |
| | | 21: Flax spot | 12221 |
| | | 31: Rice-false smut | 12231 |
| | | 41: Rice blast (leaf) | 12241 |
| | | 42: Rice blast (panicle) | 12242 |
| | | 51: Bacterial streak | 12251 |
| 13: Rape | 1: Pest | 11: Aphid | 13111 |
| | 2: Disease | 11: Sclerotinia (rhizome) | 13211 |
| | | 12: Sclerotinia (leaf) | 13212 |
| | | 21: Downy mildew | 13221 |

### 2.2. Identification Based on Neural Network

This paper presents an integrated framework of diseases and pests identification based on YOLO V3 [26]. The YOLO algorithm has been optimized and iterated, and it is better than Single Shot MultiBox Detector (SSD), Faster RCNN, and other algorithms in detection performance. In terms of network structure, YOLO V3 integrates YOLO V2, Darknet-19, and other new residual networks, composed of continuous $3 \times 3$ and $1 \times 1$ convolutional layers. The YOLO V3 network preprocesses the remote sensing image, scales it to $416 \times 416$, and sends it to the convolutional neural network for inference. Then, the non-maximum suppression (NMS) and classification recognition of the detection results are carried out according to the confidence of the network model. The YOLO V3 network divides the input image into $S \times S$ grids. If there is a target object in the center of a grid, this grid is responsible for predicting the object. Each grid predicts the four offset coordinates of the bounding box of the detected object and the confidence score. The YOLO V3 network makes predictions on three different scales, namely $13 \times 13$, $26 \times 26$, and $52 \times 52$. Three bounding boxes are predicted for each scale, and the network structure is shown in the Figure 2.

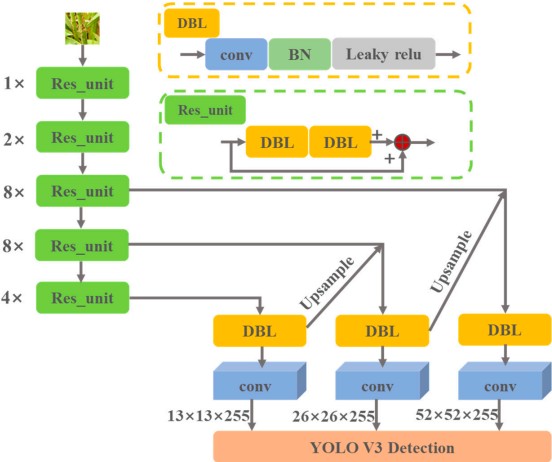

**Figure 2.** Structure diagram of Yolo V3. DBL: Darknetconv2d_BN_Leaky, BN: Batch Normalization, conv: convolution.

First, the input image is divided into $13 \times 13$ cells. Then, each cell will generate detection boxes, which are each composed of a five-dimensional prediction parameter, including the center point coordinate $(x, y)$, width and height $(w, h)$, and confidence score $s_i$. The confidence score is calculated as

$$s_i = P(C_i|O_{object}) \times P(O_{object}) \times IOU(k_T, k_p) \qquad (1)$$

where $P(C_i|O_{object})$ represents the conditional probability of predicting class $i$ in the cell when there are diseases and pests in the detection frame; and $P(O_{object})$ indicates the possibility of diseases and pests in the detection box of the current cell. If there are diseases and pests, the value is 1; otherwise, the value is 0. $IOU(k_T, k_p)$ is the intersection and union ratio of prediction detection frame and real annotation frame.

Then, the NMS algorithm is used to retain the target detection box with a high confidence score. The NMS algorithm formula is

$$s_i = \begin{cases} s_i & IOU(M, b_i) < N_i \\ 0 & IOU(M, b_i) \geq N_I \end{cases} \qquad (2)$$

where $M$ is the detection box with the highest confidence score in the current area; $IOU(M, b_i)$ is the intersection and union ratio of $M$ and the adjacent overlapping frame $b_i$; $N_I$ is the set overlap threshold.

In this paper, the loss function $CIOU$ is used to calculate the regression loss value of the prediction frame, and the formula is as follows:

$$Loss_{CIOU} = 1 - IOU + \frac{\rho^2(a, a^{gt})}{b^2} + \alpha v \qquad (3)$$

$$v = \frac{4}{\pi^2}\left(\arctan\frac{w^{gt}}{h^{gt}} - \arctan\frac{w}{h}\right) \qquad (4)$$

$$\alpha = \frac{v}{1 - IOU + v} \qquad (5)$$

where $\alpha, a^{gt}, w, w^{gt}, h, h^{gt}$ are the center position coordinates, width, and height of the prediction box and the dimension box, respectively; $\rho$ represents the Euclidean distance between the centers of the two frames; $b$ represents the diagonal length of the smallest rectangle containing the prediction box and dimension box; $\alpha v$ represents the penalty item including the width and height of the prediction box and the dimension box; $\alpha$ is the weight parameter; and $v$ represents the calculation item including the width and height of the prediction box and dimension box.

After using the loss function $\alpha, a^{gt}, w, w^{gt}, h.h^{gt}$, the optimization direction can still be given when the overlapping area of the prediction box and the annotation box is equal, disjoint, or included. This makes the prediction box close to the center of the annotation box and speeds up the convergence speed of network training.

Farmers would use mobile phones and other terminal devices to take pictures or videos and send them to the server through the Wi-Fi. After obtaining the pictures, the server uses the identification network proposed in this section combined with the expert database system to return the diseases and pests information and solutions to farmers.

### 2.3. Data Reporting and Location Information Acquisition

In this section, Geographic Information System (GIS) technology is further used to establish a geographic information system database to display the occurrence degree, transmission, and geographical distribution of crop diseases, pests, and weeds in the form of a map. Through the collected meteorological, crop varieties, pest information, control measures, and other information, combined with the mathematical model in the later stage, the temporal and spatial dynamics and law of pest and grass occurrence are

analyzed. Finally, the occurrence trend of diseases and pests are predicted by evaluating the environmental and influencing factors.

## 3. Hardware System Development and Application

As shown in Table 2, the product type is selected according to the characteristics of crop objects. The macro mode is suitable for photographing diseases and pests with a very small size; The handheld probe mode is suitable for shooting close to the photographer; The support frame mode is suitable for shooting in a far and high range from the photographer.

**Table 2.** Hardware development and composition under different modes.

| Form<br>Pattern | Wireless<br>Lens | Probe Rod Kit | Intelligent<br>Information Terminal | Temperature and<br>Humidity Sensor | Support<br>Frame | Macro<br>Lens |
|---|---|---|---|---|---|---|
| Macro mode | √ | | √ | √ | | √ |
| Handheld probe mode | √ | √ | √ | √ | | |
| Support frame mode | √ | √ | √ | √ | √ | |

Note: The probe rod kit consists of a hand-held probe rod, a vientiane folding rod, a lens connecting seat, a temperature and humidity sensor fixing seat, and an intelligent information terminal-fixing frame.

### 3.1. Perception Module Construction

As shown in Figure 3, the hardware system mainly includes temperature and humidity Bluetooth sensor, intelligent information terminal, wireless lens, and other firmware.

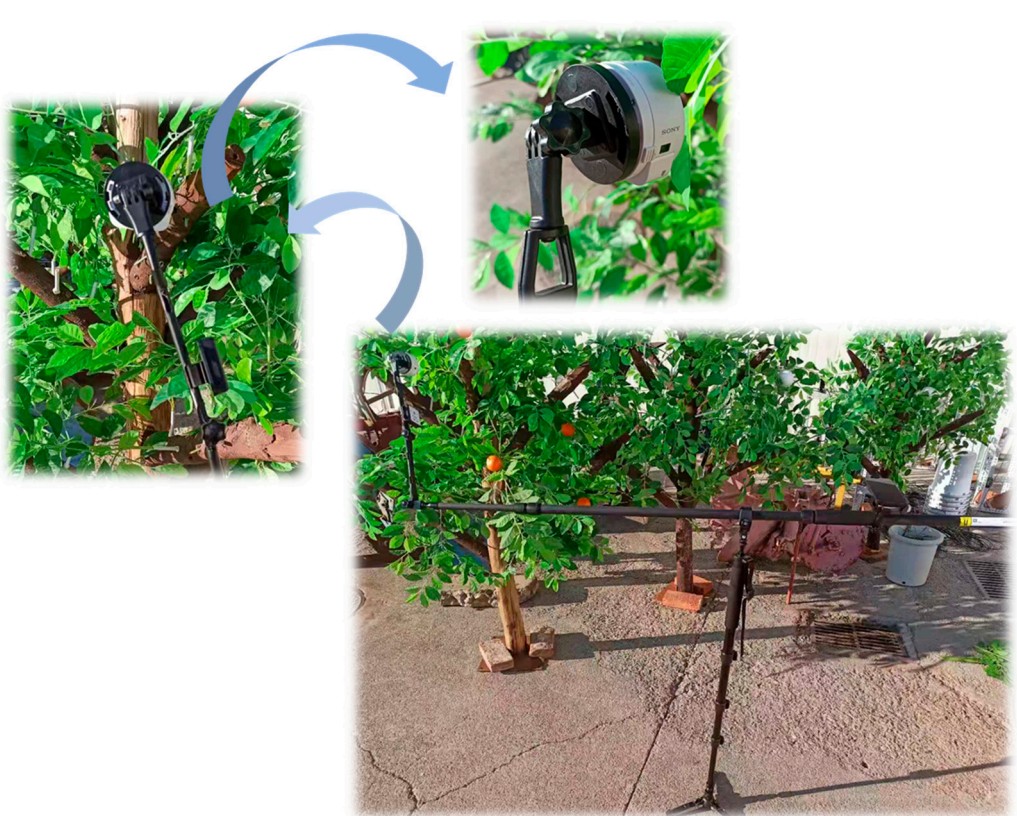

**Figure 3.** Hardware system module.

The temperature and humidity sensor measures the temperature and humidity information, connects with the intelligent information terminal through Bluetooth, and transmits information to the terminal to realize the real-time environmental information recording of farmland. The system adopts a waterproof and dustproof grade not lower than IP65 and has good drop resistance. The temperature measurement range is −20 °C to 50 °C, and the humidity is 0–100%. The detection accuracy of the temperature and humidity sensors are: humidity ±0.5% and temperature ±0.5 °C.

The intelligent information terminal includes a microprocessor, data memory, satellite positioning module, Wi-Fi communication module, Bluetooth communication module, power module, and display module. The Android version is not less than 6.0, and the rear camera is not less than 10 million pixels. It supports 4G wireless communication network communication, and the storage memory is 16G. The required image of plant diseases and pests can be clearly collected through the camera and supporting wireless camera carried by the intelligent information terminal and can be transmitted to the network service background together with GPS information through a communication protocol for further processing. The image shooting supports real-time shooting and real-time viewing.

The wireless lens realizes the front-end HD image acquisition and transmits it to the intelligent information terminal for data processing through Wi Fi, which can adapt to the harsh working environment of high temperature and humidity. The camera is 10 million pixels, supports single autofocus, and can clearly shoot single diseases and pests with a size of 1 mm–10 cm.

The telescopic probe rod can be extended to a maximum length of 2.5 m to ensure that it can collect pest images and climate information within 2.5 m away from the operator. At the same time, the long probe rod can ensure that the collection range of pests cannot be disturbed by human beings, and the collection point can represent the characteristics of typical farmland.

### 3.2. Positioning and Communication Module

The system positioning information includes the time, longitude, latitude, and other positioning status information corresponding to the collected image. The hardware developed in this paper can store the positioning information in the terminal and upload it to the monitoring center through wireless communication.

System communication modes include communication between camera, sensor, and intelligent terminal and communication between intelligent terminal and back-end service platform. The camera and the intelligent information terminal can be connected together to transmit the collected images; the environmental sensor can be connected with the intelligent information terminal to transmit the collected environmental data; and the intelligent information terminal and the back-end service platform transmit data using a wireless network. The camera communicates with the intelligent information terminal through a Wi-Fi network; the environmental sensor communicates with the intelligent information terminal through a Bluetooth network; and the intelligent information terminal communicates with the network service platform through 3G and 4G networks. The terminal sends pictures, environmental data, and other relevant data to the network service platform for analysis and processing through TCP/IP protocol.

## 4. Software System Development

According to the different needs of users, the software system developments of two platforms are carried out in this work. The software can be obtained through the website (www.zkzngz.com, accessed on 20 April 2022). At this stage, it is only available in Chinese. The team is developing versions in English and other languages to provide users in different regions.

### 4.1. Development of Mobile Terminal Software System

The terminal adopts a Wi-Fi connection with a wireless lens and sensor, and the system interface form is shown in Figure 4a. By clicking the environmental sensor and camera that need to be linked, the link between the mobile phone and device can be realized so as to establish wireless transmission. In the system design, the intelligent identification of plant diseases and pests is mainly divided into four steps: wireless shooting, image uploading, intelligent identification, and system storage. In the process of wireless shooting, the plant phenotype can be obtained at a macro distance through the connected wireless lens and the operation of retracting and stretching the lens, as shown in Figure 4b. After shooting,

the obtained plant image is recognized by jumping the network link. In the recognition process, one or more pictures can be recognized continuously, and crop type and growth period need to be selected at the same time. The software system can also record the history of identification. At the same time, the recognition result of the collected image will be sent back to the user, and the solution of this kind of pest will be given.

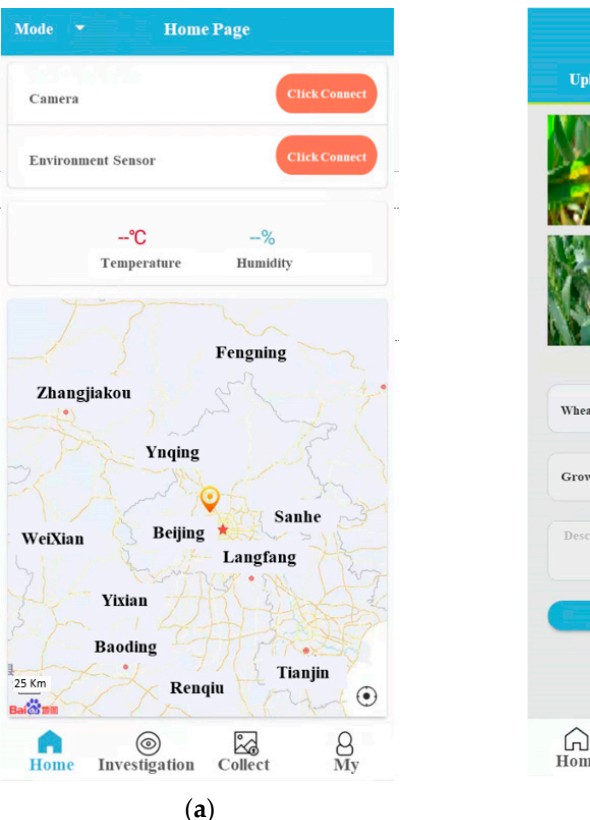
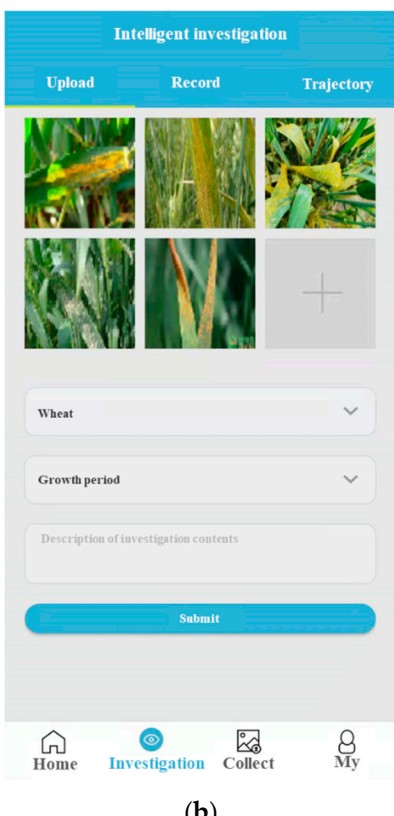

(**a**)　　　　　　　　　　　　　　　(**b**)

**Figure 4.** Mobile terminal interface. (**a**) The main interface displays geographical location and meteorological data. (**b**) Image upload and selection of crops and growth period.

### 4.2. Development of Computer Terminal Software System

In order to record and analyze various diseases and pests in the field, it is necessary to set up multiple mobile monitoring systems in different regions and record and count the disease and pest information and environmental data obtained by the mobile monitoring system. The project has further developed the computer software system, which can accept and record the data of multiple mobile monitoring systems and display the regional location, intensity, and trend of diseases and pests.

## 5. Experiment and Discussion

### 5.1. Image Collection of Diseases and Pests

When using the mobile intelligent device for image acquisition, the user should see a relatively clear picture on the screen (i.e., the lens focusing is completed) and then click the shooting button to avoid dithering and fuzzy image data.

In the test experiments in this section, there were six kinds of common diseases and pests: rice planthopper, rice-leaf roller, wheat spider, wheat aphid, wheat scab, and sheath blight. For the area where each pest of rice or wheat gathers on the plant, the area where the disease occurs on the plant, and the significant characteristic area of different seedling conditions, the camera shall aim at the characteristic area for multi-angle shooting (such as side, top, middle, local, etc.), and the specific shooting standards are shown in Table 3.

**Table 3.** Collection mode under different crops with diseases and pests.

| Type | Name | Viewfinder Range | Angle and Distance |
|---|---|---|---|
| Pest | Rice planthopper | The side is facing the aggregation area of rice planthopper above the rice root. The lens vertically penetrates into the cluster and overlooks the aggregation area of rice planthopper above the root. | Angle: 0~30°. Distance: 5~10 cm, 10~15 cm, mainly in a clear state with a distance of 5~10 cm. |
| | Rice-leaf roller | The viewfinder shall mainly focus on the side images of rice leaves, which can be distinguished by the naked eye. | Angle: 30~60°. Distance: 10~20 cm, 20~30 cm, mainly in a clear state with a distance of 20~30 cm. |
| | Wheat spider | The lens is close to the ground in the winter and perpendicular to the ground in the spring. | Angle: 0~90°. Distance: 5~10 cm, 10~15 cm, mainly in a clear state with a distance of 5~10 cm. |
| | Wheat aphid | The whole wheat plant shall be photographed before jointing, and the middle and upper parts of the wheat plant shall be photographed after booting. | Angle: 0~30°. Distance: 5~10 cm, 10~15 cm, mainly in a clear state with a distance of 5~10 cm. |
| | Others | Other pests, such as striped borer, leaf cicada, rice borer sandfly, and armyworm, can be photographed according to the actual situation of the investigation. | According to the survey characteristics, refer to the above form. |
| Disease | Sheath blight | The middle and lower part of rice is photographed (one side is used). | Angle: 0~30°. Distance: 10~20 cm, 20~30 cm, mainly in a clear state with a distance of 10~20 cm. |
| | Leaf rust | The viewfinder shall mainly focus on the side image of wheat ear (which can be distinguished by naked eyes). | Angle: 30~60°. Distance: 10~20 cm, 20~30 cm, mainly in a clear state with a distance of 20~30 cm. |
| | Others | Other diseases, such as bacterial stripe disease, rice false smut, and ear neck blast, can be photographed according to the actual situation of the investigation and with reference to the above types. | According to the survey characteristics, refer to the above form. |
| | Seedling growth | According to the significant characteristic areas of different growth periods, the viewfinder should mainly focus on the side image of crop seedling. | Angle: 30~60°. Distance: 10~20 cm, 20~30 cm, mainly in a clear state with a distance of 20~30 cm. |

*5.2. Analysis of Acquisition Results*

The six most common diseases and pests (rice planthopper, rice-leaf roller, rice sheath bright, wheat spider, wheat aphid, and wheat leaf rust) were selected for statistical analysis by different methods. The sample size of each kind of diseases and pests was 200, and the rice images were collected in Liyang, Jiangsu, China (119.67° E, 31.49° N); Huaiyuan, Anhui, China (117.04° E, 32.97° N); and Changsha, Hunan, China (113.33° E, 28.05° N). The wheat images were collected in Xuchang, Henan, China (113.77° E, 34.05° N); Baoding, HeBei, China (115.28° E, 38.89° N); and Linyi, Shandong, China (118.259° E, 35.19° N). The collection method is shown in Table 3. In the experiment, the average precision (*AP*) was used to evaluate the effects of different methods. The results are shown in Table 4. The *AP* is defined as

$$AP = \int_0^1 P(R)dR \qquad (6)$$

$$R = \frac{TP}{TP + FN} \tag{7}$$

$$P = \frac{TP}{TP + FP} \tag{8}$$

where *AP* is the average detection accuracy, *R* is the recall rate, *P* is the accuracy, *TP* represents the positive samples predicted as positive, *FN* represents the positive samples predicted as negative, and *FP* represents the negative samples predicted as positive.

**Table 4.** Average precision (AP) of different methods on different diseases and pests.

|  | Rice Planthopper | Rice-Leaf Roller | Rice Sheath Blight | Wheat Spider | Wheat Aphid | Wheat Leaf Rust |
|---|---|---|---|---|---|---|
| Ours (Yolo V3) | 83.32% | 85.34% | 82.03% | 86.32% | 85.53% | 87.01% |
| Yolo V2 [27] | 78.13% | 80.78% | 78.89% | 81.32% | 80.56% | 83.65% |
| Faster RCNN [28] | 75.52% | 77.45% | 76.76% | 75.89% | 73.20% | 79.32% |
| SSD [29] | 72.32% | 75.32% | 77.27% | 74.12% | 71.32% | 74.78% |

As shown in Table 4, the Yolo V3 method used in this paper had a higher accuracy than other detection methods for different diseases and pests. Yolo V3 had an average improvement of 10% compared with SSD in the detection of all diseases and pests; Yolo V3 had an average improvement of 5% compared with Yolo V2. The experimental statistical results show that Yolo V3 had a better perception and learning ability for diseases and pests images. The statistical methods and identification effects for different diseases and pests are as follows.

### 5.2.1. Rice Planthopper

The monitoring of rice planthopper should be conducted once every five days after the transplanting field returns to green. The direct seeding field starts from 30 days after rice sowing until the end of rice yellow maturity. During the experiment, two rows were chosen on each side of each field for a total of eight rows, one meter per row, and a total of eight meters. The front end of the camera of the mobile acquisition terminal went deep into the rice bush. The lenses were 5 cm–10 cm away from the rice base and aimed at the middle and lower part of the rice. One picture was taken at an interval of 20 cm, five pictures were taken in each line, and forty pictures were taken in each field. The number of shots per line can also be adjusted according to the actual situation. In principle, there is no remake or missed shot, or one shot per cluster. As shown in Figure 5, the server intelligently identifies the rice planthopper and counts it automatically.

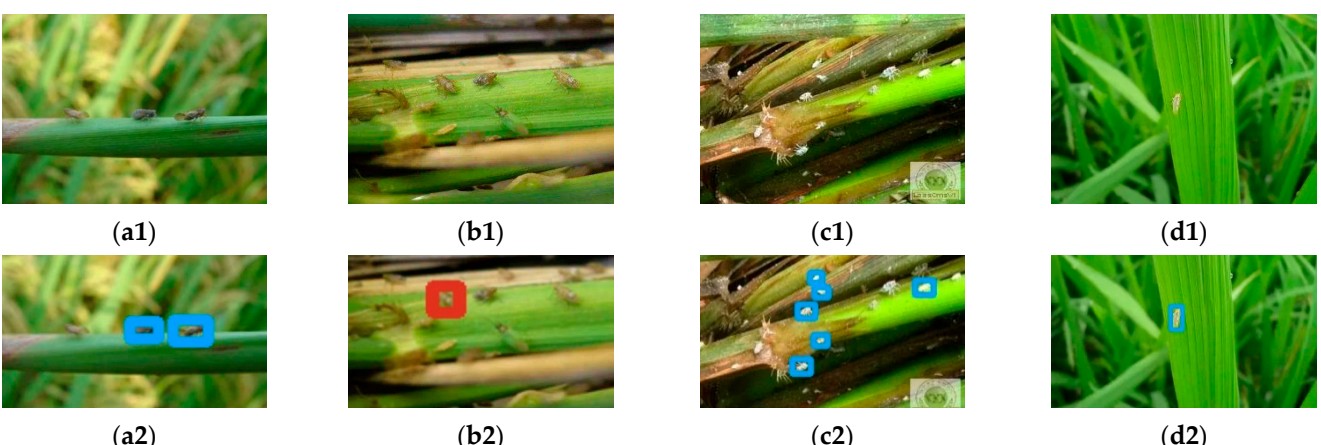

**Figure 5.** Rice planthopper. (**a1**), (**b1**), (**c1**), and (**d1**) are the original images; (**a2**), (**b2**), (**c2**), and (**d2**) are the recognition results.

### 5.2.2. Rice-Leaf Roller

The monitoring of the rice-leaf roller begins with adults in the field, driving moths, and taking photos day by day. Three representative paddy fields were selected and investigated. The researchers caught moths every morning, checked 10 m along the edge of each field, walked against the wind, slowly moved the rice plant with a pole, and counted the number of rice-leaf roller moths flying at the same time. The specific experimental operations were as follows: (1) Pest collection: one person drives moths, and the other takes photos immediately after them. When the person in front drives every square meter of rice, and the adults fly up, the person behind takes photos immediately. (2) Collection of leaf curl rate: five points are randomly sampled from each field; more than one third of the rice leaves are photographed at each point; and the damage status of the rice-leaf roller is photographed. The identification results are shown in Figure 6.

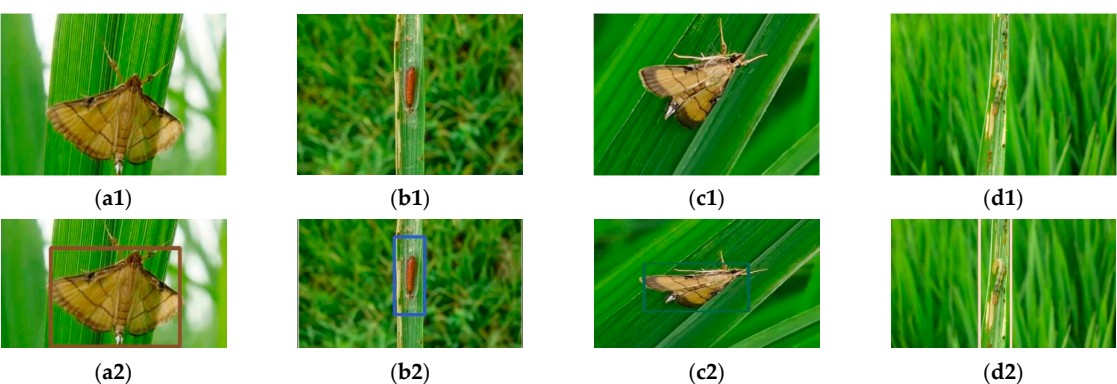

**Figure 6.** Rice-leaf roller. (**a1**), (**b1**), (**c1**), and (**d1**) are the original images; (**a2**), (**b2**), (**c2**), and (**d2**) are the recognition results.

### 5.2.3. Rice Sheath Blight

The monitoring of sheath blight began at the tillering stage of rice and was investigated every five days until the end of the milk ripening of the rice. Three representative types of fields were selected for the experiment. The four sides of each field were 2 m away from the side of the field. Two rows were taken along the ridge and rice-transplanting row. Each length was 1 m, and a total of 8 m was checked. The camera of the mobile acquisition terminal went deep into the rice bush; the lower end of the camera was 10–20 cm away from the rice base; and the camera was aimed at the middle and lower parts of the rice (one-side shooting was adopted). One photo was taken every 20 cm; five photos were taken in each line; and forty photos were taken in each field. The recognition results are shown in Figure 7.

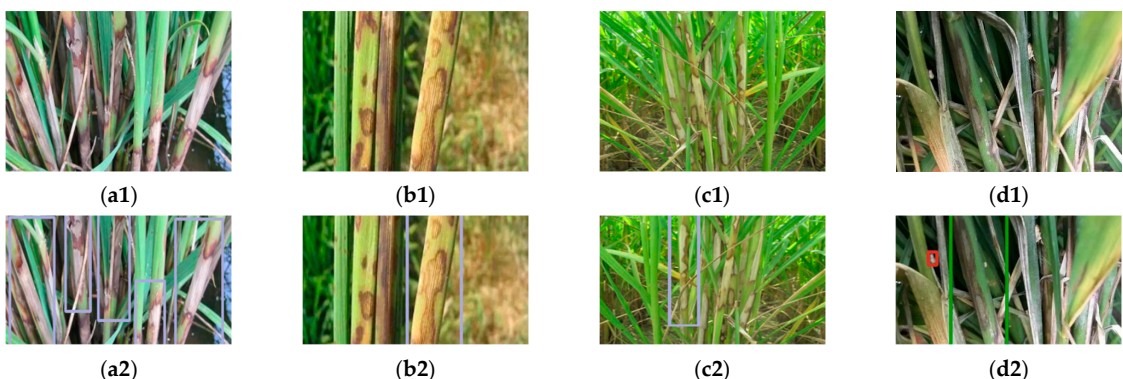

**Figure 7.** Rice sheath blight. (**a1**), (**b1**), (**c1**), and (**d1**) are the original images; (**a2**), (**b2**), (**c2**), and (**d2**) are the recognition results.

### 5.2.4. Wheat Spider

Wheat spiders need to be monitored in the winter and spring. The collection was carried out every day from 2:00 p.m. to 3:00 p.m. in the winter and from 3:00 p.m. to 4:00 p.m. in the spring, once every five days. Three representative wheat fields of different types were selected, and the area of each field was not be less than 667 m². Two persons were required to cooperate in each data acquisition, one taking photos and one investigating data. Each field was photographed along a row of wheat in the middle of the field. The first photographing point was 2 m away from the field head. Later, 1 photo was taken every 2 m, and a total of 20 photos was taken. When taking pictures, the camera clung to the right ridge and took pictures to the left ridge. The lens was attached to the ground in the winter and 5 cm–10 cm away from the ground in the spring (the lens faces the wheat). When taking photos with mobile intelligent acquisition equipment, the lens was placed vertically. The collected pictures and recognition results are shown in Figure 8.

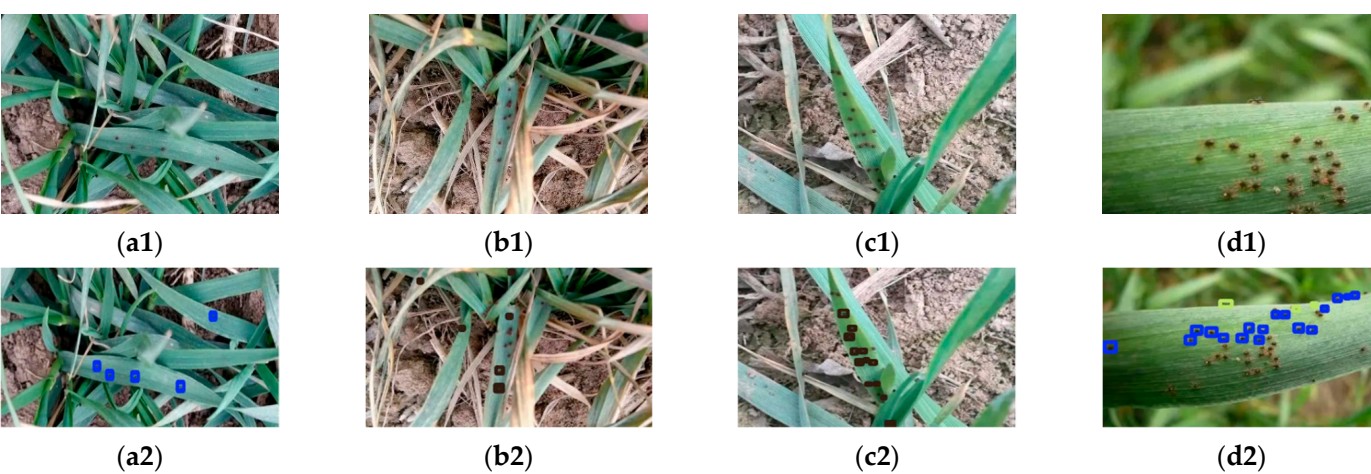

**Figure 8.** Wheat spider. (**a1**), (**b1**), (**c1**), and (**d1**) are the original images; (**a2**), (**b2**), (**c2**), and (**d2**) are the recognition results.

### 5.2.5. Wheat Aphid

The investigation was to be conducted once every seven days from the green jointing stage to the milk ripening stage of wheat. When the number of aphids rose sharply and the daily increase of aphids exceeded 300, the investigation was to be conducted once every five days. More than 10 representative wheat fields were selected according to sowing date, variety, growth, and other conditions. Samples were taken at five diagonal points of each field; 50 plants were investigated at each point before jointing; and 20 plants were investigated at each point after booting. The lenses were 5 cm–10 cm away from the wheat plant, close to the right ridge, and took photos to the left ridge. The whole wheat plant was to be photographed before jointing, and the middle and upper parts of the wheat plant were to be photographed after booting. The 5 photos were to be photographed for each field, and more than 50 photos were to be photographed for each survey. The collected pictures and recognition results are shown in Figure 9.

### 5.2.6. Wheat Leaf Rust

The investigation was conducted every five days from the jointing stage of wheat to the end of milk maturity of wheat. Each field was sampled in parallel at five points, and two pictures were taken at each point with an interval of 1 m for a total of 10 m. The lens of the equipment was aimed at more than one third of the wheat for photographing. The collected pictures and recognition results are shown in Figure 10.

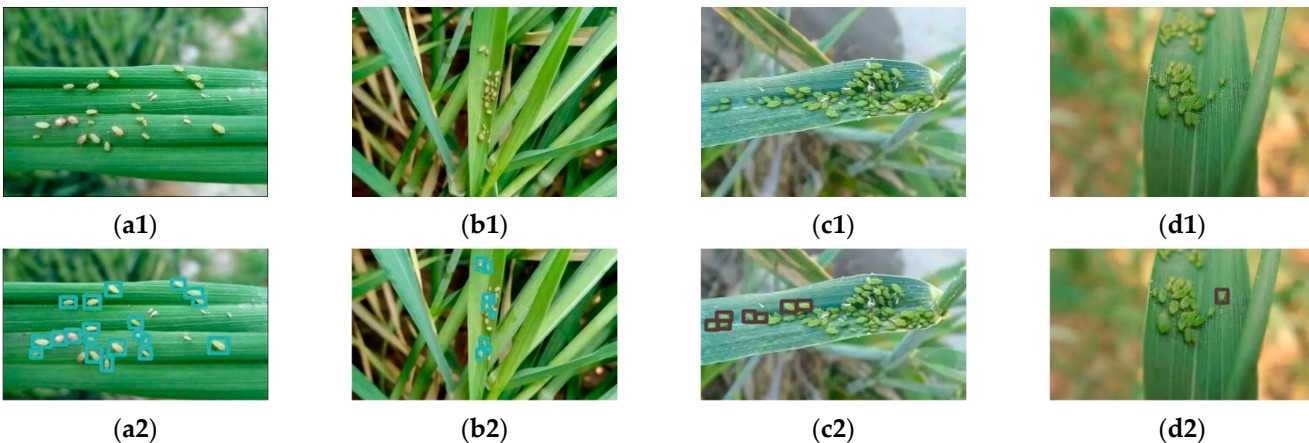

**Figure 9.** Wheat aphid. (**a1**), (**b1**), (**c1**), and (**d1**) are the original images; (**a2**), (**b2**), (**c2**), and (**d2**) are the recognition results.

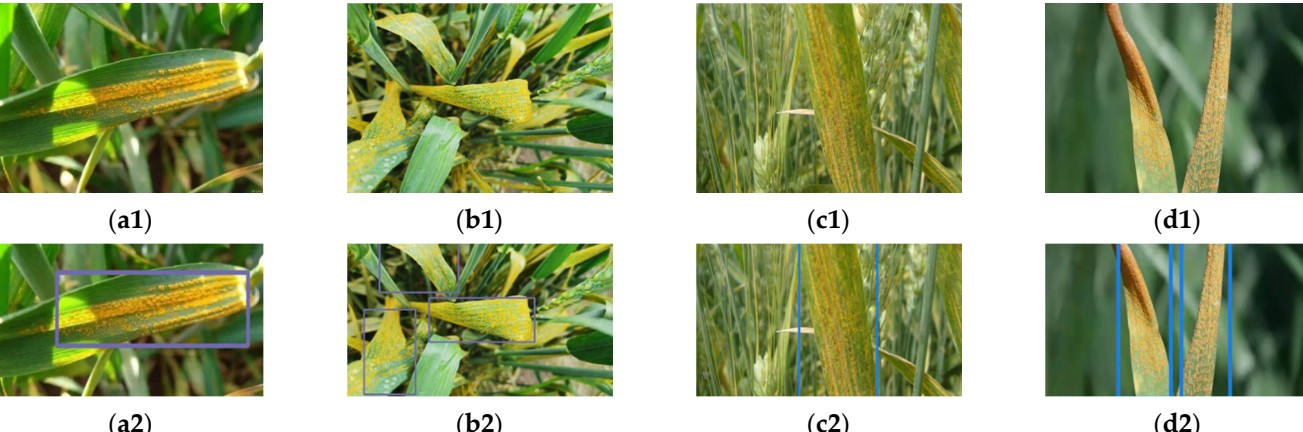

**Figure 10.** Wheat leaf rust. (**a1**), (**b1**), (**c1**), and (**d1**) are the original images; (**a2**), (**b2**), (**c2**), and (**d2**) are the recognition results.

## 6. Conclusions

The intelligent pest detection kit serves plant protection management through "sensing + artificial intelligence" technology to realize the accurate collection and mobile monitoring of plant information and data. The equipment has been loaded with more than 500 users in Qinghai, Jiangxi, Anhui, Zhejiang, Shanxi, and other places in China. The suite provides users with tools and means for plant protection image and microenvironment acquisition to help users implement more-refined plant protection management. Through artificial intelligence to identify diseases and pests of crops, the plant protection operation is easier, more accurate, and more efficient, so anyone can have a plant protection expert with him. At this stage, this project only identifies the main diseases and pests of wheat, corn, rice, and other major field crops and continues to increase the sample types to realize more crop disease and pest monitoring, which is the further research direction of intelligent agricultural monitoring.

**Author Contributions:** Conceptualization, S.W. and P.Q.; methodology, S.W.; software, S.W.; validation, S.W., P.Q., and X.H.; formal analysis, S.W.; investigation, S.W.; resources, W.Z.; data curation, W.Z.; writing—original draft preparation, S.W.; writing—review and editing, S.W.; visualization, S.W.; supervision, S.W.; project administration, X.H.; funding acquisition, X.H. All authors have read and agreed to the published version of the manuscript.

**Funding:** This research was funded by the Professor workstation of intelligent plant protection machinery and precision pesticide application technology, grant number 202005511011020; the

National Modern Agricultural Industrial Technology System of China (No. CARS-28-20); Deutsche Forschungsgemeinschaft (DFG, German Research Foundation)-328017493/GRK 2366.

**Institutional Review Board Statement:** Not applicable.

**Informed Consent Statement:** Not applicable.

**Data Availability Statement:** Not applicable.

**Acknowledgments:** We would like to thank Lijun Zhou and Jian Wang for their helpful work on their preliminary basic work.

**Conflicts of Interest:** The authors declare no conflict of interest.

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
