# Peer review of "Development and Application of an Intelligent Plant Protection Monitoring System"

_agronomy, doi:10.3390/agronomy12051046_

Round 1

Reviewer 1 Report

Fig.1. more description required 

Still English language improvement is required in the experimental part. 

Equations need to be cite 

Author Response

Response:

Thanks for pointing it out. We give a supplementary description of Figure 1. We improved the language of the experimental part. The key equations of Yolo V3 are cited, and three key equations are cited in the evaluation results. We sincerely hope to get your approval.

Above statements are our responses to the comments from reviewer. The authors would like to acknowledge the editor and reviewer again for their great contributions and efforts for improving this paper.

Reviewer 2 Report

The authors referred to the comments from the previous review for the manuscript. I accept changes. In the future, I suggest using more precise  describing relationships between the parameters studied. They supplemented the discussion which strengthens the message and importance of information in the manuscript.

Author Response

Response:

I sincerely thank the expert for your previous contributions. Thank you very much for your suggestions. Indeed, we should further strengthen theoretical research and explore the relationship between parameters and phenomena according to your suggestions in the next step. We also hope that the next research results can be reviewed by you again.

The authors would like to acknowledge the editor and reviewer again for their great contributions and efforts for improving this paper.

Reviewer 3 Report

This is a re-submitted version of the manuscript (another journal within the same publisher). As I have already reviewed the two previous versions, I am already familiar with the content. The problems related to the previous submissions have now been resolved. All my comments from the previous round have been taken into account. This allows me, finally, to recommend the acceptance of the paper.

Author Response

Response:

Thank you for your previous reviews and suggestions. I sincerely thank the expert for your previous contributions. Best wishes to you.

This manuscript is a resubmission of an earlier submission. The following is a list of the peer review reports and author responses from that submission.

Round 1

Reviewer 1 Report

The submitted manuscript proposes an intelligent plant protection monitoring system with digital image processing. The article is in general well written, but some more details from computer science are missing. The illustrated application shows the benefits of using the systems for the farmers. Below, I list point by point my comments, both major and minor (ordered according to the occurrence in the text, not according to priority):

Line 13: “this paper develops” -> the paper does not develop a system, it is better to write “this paper proposes”.

Lines 18-19: “at any time” written twice.

Line 19: which kind of questions? Please give a short example in the abstract.

Lines 26-38: the introduction should be supported by appropriate references.

Line 50: what is ML? (all shortcuts should be defined when used for the first time)

Line 57: 582.1 K of what?

Lines 65-66: are these names of the methods?

Line 79: what is PLZ?

Line 113: it would be better to add horizontal lines between groups (pest / disease).

Line 115 (and the whole subsection): any information about the identification/detection (training, test data, etc.) is missing.  

Line 117: what is SDD?

Lines 119-121: rephrase the sentence (four times “and”).

Line 125: numbers lower than ten should be written in words (check also other parts of the manuscript).

Line 128: what are the components in the figure (e.g. Leaky relu)?

Line 193: humidity is not measured in Celsius grads.

Line 233: rephrase the sentence. Now it looks like an advertisement. The site is available only in Chinese, it should be mentioned in the manuscript.

Line 256: figure 5 is redundant and does not bring anything.

Line 312: should be 2 p.m. (not 14 p.m.), etc.

Line 351: comparing table 1 and table 3: so far only some problems can be identified. Why these pests and diseases have been chosen (table 3)? Why are the other (from table 1) not detected?

Line 393: missing journal name and year.

Line 400: missing authors and journal name.

There are also some English grammar and typing errors. The whole manuscript should be carefully checked.

Author Response

Dear expert:

Many thanks for the valuable comments on the manuscript which are much appreciated. We have revised the paper according to these constructive comments. The detailed response to each comment is given below. We wish our sincere efforts would satisfy the requirements.

Response (Line 13): Thanks for pointing it out. In the new version, we have revised it according to the suggestions of expert.

Response (Line 18-19): Thanks for pointing it out. We deleted the superfluous content.

Response (Line 19): Thanks for pointing it out. In the new version, we give a short example in the abstract.

Response (Line 26-38): We added new references in line 26-38.

Response (Line 50): Thanks for pointing it out. In the new version, we have revised it according to the suggestions of expert.

Response (Line 57): 582.1 K is 582.1×1000. K means “kilo-”. In order to reduce ambiguity, the standard counting method is adopted in the new vision.

Response (Line 65-66): Yes, these are names of the methods?

Response (Line 79): PTZ is Pan Tilt Zoom. In the new version, we have defined it.

Response (Line 113): Thanks for pointing it out. The expert's proposal is very helpful. we added horizontal lines between groups. After adding lines, the table is more clear and logical.

Response (Line 115): Thanks for pointing it out. We have revised it according to the suggestions of expert, and added information about the identification/detection. Table 3 is added to describe in detail the method of collecting pictures (including angle and distance). In Section 5.2, the analysis of the time and quantity of image acquisition is added. In Section 3.1, the size and pixel description of the acquired image are added.

Response (Line 117): SSD (Single Shot MultiBox Detector) is target detection algorithm. In the new version, we have defined it.

Response (Line 119-121): Thanks for pointing it out. We have rephrased the sentence.

Response (Line 125): Thanks for pointing it out. In the new version, we have revised it according to the suggestions of expert.

Response (Line 128): Thanks for pointing it out. Figure 2 shows the structure of Yolo v3. In fact, the whole article does not describe Yolo V3 in detail, but uses V3 network according to our actual application. If the composition of V3 is described in detail, it will take a lot of space. If readers need further understanding, they can check the following literature.

[] Redmon J , Farhadi A . YOLOv3: An Incremental Improvement[J]. arXiv e-prints, 2018.

Response (Line 193): Thanks for pointing it out. In the new version, we have revised it.

Response (Line 233): Thanks for pointing it out. The expression in the original text is indeed not standardized. We rewrite this sentence. In addition, our software language is still Chinese at this stage. We are developing an English version for users in different regions. In the manuscript, we made a supplementary explanation.

Response (Line 256): Figure 5 is computer terminal interface. It can accept and record the data of multiple mobile monitoring systems, and display the regional location, intensity and trend of diseases, pests. Therefore, we have listed the charts for display, and we hope the experts will approve it.

Response (Line 312): In the new version, we have revised it according to the suggestions of expert.

Response (Line 351): Thanks for pointing it out. Table 3 is a part of Table 1. In our test, we actually carried out all experiments on table 1. Table 3 only selected the more representative diseases and pests for detailed description and listed the experimental results.

Response (Line 393): Thanks for pointing it out. In the new version, we have revised it.

Response (Line 400): Thanks for pointing it out. In the new version, we have revised it.

Thank the expert for your suggestions. We check the manuscript again and correct the errors and typos, and we also invited native speakers to check and polish the manuscript. I hope this version can satisfy you.

Above statements are our responses to the comments from reviewer. The authors would like to acknowledge the reviewer again for the great contributions and efforts for improving this paper.

Reviewer 2 Report

The title of this study is: Development and application of an intelligent plant protection monitoring system. In this work, the authors wish to develop a variety of crop diseases and pests monitoring system taking into account meteorological conditions, plant images and expert system.

The Introduction to the study is too brief and does not end with a clearly stated purpose. The Methodology section provides the reader with enough information to repeat the experiments conducted. Information on the method of statistical analysis of the work results was not included. Image analysis allows to obtain a description of the examined image based on the obtained parameters, e.g. shape, size, color, color, etc. The obtained parameters can then be subjected to statistical analysis in order to determine the dependence or significance of the obtained results. No statistical analyzes were performed here. For the most parts the 4 and 5 section is well structured. The Discussion is no full comparison and confrontation with the research of other authors in this area. The Conclusions chapter contains information obtained after conducting experiments but there were no comparison and confrontation with the research of other authors in this area.

Part: References.

The literature used is appropriate but should be supplementing about the items from the last years of publication about similar problem. The literature should be supplemented with additional items describing the examined aspects.

Author Response

Dear expert:

Many thanks for the valuable comments on the manuscript which are much appreciated. We have revised the paper according to these constructive comments. The detailed response to each comment is given below. We wish our sincere efforts would satisfy the requirements.

Thanks for pointing it out. In the new version, we have revised it according to the suggestions of expert.

In the Introduction part, We further supplemented the references and the research status. The references take the ground equipment and air equipment as the review direction, list in detail the work done by scholars in the crop diseases and pest monitoring system in recent years, and comment on the advantages and disadvantages of its methods. Finally, we introduce our research content and the main contributions, and give the overview of our research in the form of picture (Figure 1). Thank the expert for your suggestions. We sincerely hope to get your approval.

In the Experiment and Discussion part, Table 3 is added to describe in detail the method of collecting pictures (including angle and distance). In Section 5.2, the analysis of the time and quantity of image acquisition is added. In Section 3.1, the size and pixel description of the acquired image are added. Thank the expert for your suggestions. We sincerely hope to get your approval.

In the Conclusions part, we added comparison and discussion with other researchers, and cited the literature of other authors. Thank the expert for your suggestions. We sincerely hope to get your approval.

In the References part, we supplement the research literature in similar fields, and analyze and discuss it in the introduction. Thank the expert for your suggestions. We sincerely hope to get your approval.

Above statements are our responses to the comments from reviewer. The authors would like to acknowledge the editor and reviewer again for their great contributions and efforts for improving this paper.

Reviewer 3 Report

The authors have presented a monitoring system through different sensors to capture plant images as well as meteorological information for providing solutions to diseases and pests on field crops. It is well presented with a proper setup experiment design. However, some concerns are there for improvement and therefore, a revision has been suggested.

Comments:

  • Introduction: Wherever Reference [3, so on…] written, it should be replaced by Author names as per the usual style of the journal.
  • Fig 2: It needs description along with their abbreviation used therein (e.g. DBL, BN, conv, etc...)
  • L123: What is the input image size (dimension) into S X S grid
  • L127: Scales such as 13 × 13, 26 × 26, 52 × 52, what is the unit of these scales
  • After section 4.2: How pattern recognition has been done based on which feature inputs. The description of this part as well as the artificial intelligence part still missing in the manuscript
  • How the end solution on identifying diseases and pests reaches farmers which have to be described before the conclusion section
  • Before the conclusion section, limitations of the study can be included and future scope for precision farming.
  • Line167: Abbreviate GIS

Author Response

Dear expert:

Many thanks for the valuable comments on the manuscript which are much appreciated. We have revised the paper according to these constructive comments. The detailed response to each comment is given below. We wish our sincere efforts would satisfy the requirements.

Response (Introduction): Thanks for pointing it out. In the new version, we have revised it according to the suggestions of expert.

Response (Fig 2): Thanks for pointing it out. In the new version, we explain all abbreviations in Fig 2. Fig 2 shows the structure of Yolo v3. In fact, the whole article does not describe Yolo V3 in detail, but uses V3 network according to our actual application. If the composition of V3 is described in detail, it will take a lot of space. If readers need further understanding, they can check the following literature.

[] Redmon J , Farhadi A . YOLOv3: An Incremental Improvement[J]. arXiv e-prints, 2018.

Response (Line 123): Thanks for pointing it out. The size of S × S is not fixed. In the process of image processing, different network layers will have different sizes.

Response (Line 127): Thanks for pointing it out. The unit of these scales are in pixels.

Response (After section 4.2): Thanks for pointing it out. We added the description of the artificial intelligence in Section 2.2. Section 4.1 adds how users can get the final solution. We agree with the expert's suggestions. In the new version, in the conclusion part, we summarize the advantages and disadvantages of our method, and look forward to the next research.

Response (Line 167): Thanks for pointing it out. In the new version, we have defined it.

Above statements are our responses to the comments from reviewer. The authors would like to acknowledge the editor and reviewer again for their great contributions and efforts for improving this paper.

Reviewer 4 Report

Critical contribution is missing in the introduction section

Very hard to find the novelty in this manuscript

How does the author claim the novelty?

The authors need to mention the seed paper and demonstrate how the current work is better than seed.

Required in-depth description of the dataset used in this research 

Author Response

Dear expert:

Thank you for your comments and suggestions. The expert's question is very practical. We apologize for the missing contribution of our manuscript. In the new version, we added the contributions and organization of this paper. And other chapters are also deeply improved. In fact, the focus of the full text is to do a system integration. We have carried out in-depth development of deep learning technology, sensor technology, software and expert system, and finally formed a good product, which can be actually promoted and applied. At the same time, we also open source our software, and everyone can download and apply it for free. In fact, after nearly two years, the team members have obtained a large number of data sets and conducted multiple rounds of debugging, and finally formed this relatively mature system. Although the system is not very innovative in the theoretical field, it will be very beneficial to users in practice because of its high integration and mature software system. The team author hopes to promote the development of agriculture through this system and benefit more agricultural producers.

We sincerely hope to get your approval.

Above statements are our responses to the comments from reviewer. The authors would like to acknowledge the editor and reviewer again for their great contributions and efforts for improving this paper.

Round 2

Reviewer 1 Report

I appreciate the efforts of the authors to improve the quality of the submitted manuscript. Although the article is better now, unfortunately, some issues are still open.

Regarding action taken to the comment in line 19: The added text is not a question.
I leave this comment still open.

Regarding action taken to the comment in 261 and the sentence: “As shown in Figure 5, the project has further developed the computer software system, which can accept and record the data of multiple mobile monitoring systems, and display the regional location, intensity and trend of diseases, pests” -> the figure does not bring anything. The most part of the screenshot is empty. The authors should either present the application with some real sample data or just delete the figure.

English language should also be checked once more, see e.g. “The wireless lens realizes” -> should be “realize” (line 208).

Author Response

Dear expert:

Many thanks for the valuable comments on the manuscript which are much appreciated. We have revised the paper according to these constructive comments. The detailed response to each comment is given below. We wish our sincere efforts would satisfy the requirements.

Response (19): Thanks for pointing it out again. We have rephrased the sentence.

Response (Fig 5): Thanks for pointing it out again. After discussion with other authors, it is found that the amount of information in the original figure 5 is indeed too small, and the insertion in the text is not very meaningful. Finally, we agree with the expert's suggestion and deleted figure 5.

Response (208): Thanks for pointing it out. we have revised it. We checked the full manuscript again to avoid some clerical errors.

Above statements are our responses to the comments from reviewer. The authors would like to acknowledge the expert again for the great contributions and efforts for improving this paper.

Reviewer 2 Report

The title of this study is: Development and application of an intelligent plant protection monitoring system. Thanks for replying to my comments. Unfortunately, I have not received an answer to the most important issues that I will present below.

The title of this study is: Development and application of an intelligent plant protection monitoring system. Thanks for replying to my comments. Unfortunately, I have not received an answer to the most important issues that I will present below.

Information on the method of statistical analysis of the work results was not included again. I have not found any information on how the applied system protects plants?

This is a scientific article and should contain basic elements and even basic statistical analysis of the results obtained. This work does not include it, and the detection of objects in an image is nothing new and revealing. The authors claim that: “YOLO algorithm has been optimized and iterated, and it is better than Single Shot MultiBox Detector (SSD), Faster RCNN and other algorithms in detection performance”. No evidence was provided in the form of a comparison of the results obtained with the given algorithms.

Author Response

Dear expert:

Many thanks for the valuable comments on the manuscript which are much appreciated. We have revised the paper according to these constructive comments. The detailed response to each comment is given below. We wish our sincere efforts would satisfy the requirements.

Thanks for pointing it out again. First of all, I'm very sorry that our previous answer didn't satisfy you. Statistical analysis is indeed missing in this paper. In this round of revision, we focus on the performance of different methods in our data sets, and describe the collection locations of different data sets in detail.

This paper focuses on how to identify the types of diseases and pests and upload the temperature and humidity information to the system. It is a front-end detection system in the field of plant protection, which provides a reference basis for subsequent pesticide spraying and other operations.

Above statements are our responses to the comments from reviewer. The authors would like to acknowledge the editor and reviewer again for their great contributions and efforts for improving this paper.

Reviewer 3 Report

Authors have addressed all the previous queries. I suggest you should add at least one reference that gives details on YOLOv3.

Author Response

Dear expert:

Many thanks for the valuable comments on the manuscript which are much appreciated. We have revised the paper according to these constructive comments. The detailed response to each comment is given below. We wish our sincere efforts would satisfy the requirements.

Response: Thanks for pointing it out again. We agree with the expert's suggestion. Indeed, in order to make our manuscript more rigorous, we cited reference of Yolo V3 in Section 2.2.

Above statements are our responses to the comments from reviewer. The authors would like to acknowledge the expert again for the great contributions and efforts for improving this paper.